# Review of Related Factors for Persistent Risk of Hepatitis B Virus-Associated Hepatocellular Carcinoma

**DOI:** 10.3390/cancers16040777

**Published:** 2024-02-14

**Authors:** Nevin Varghese, Amry Majeed, Suraj Nyalakonda, Tina Boortalary, Dina Halegoua-DeMarzio, Hie-Won Hann

**Affiliations:** 1Department of Medicine, Thomas Jefferson University Hospital, Philadelphia, PA 19107, USA; nevin.varghese@jefferson.edu (N.V.); amry.majeed@jefferson.edu (A.M.); suraj.nyalakonda@jefferson.edu (S.N.); tina.boortalary@jefferson.edu (T.B.); dina.halegoua-demarzio@jefferson.edu (D.H.-D.); 2Division of Gastroenterology and Hepatology, Department of Medicine, Thomas Jefferson University Hospital, Philadelphia, PA 19107, USA

**Keywords:** hepatitis B virus, hepatocellular carcinoma, nucleoside analog, antiviral therapy, hepatitis cure, cccDNA

## Abstract

**Simple Summary:**

Hepatitis B virus (HBV) affects around 300 million people worldwide and is a significant risk factor for the development of hepatocellular carcinoma (HCC). Nucleos(t)ide analog therapy has aided in decreasing mortality from HBV. However, no cure for HBV currently exists. Despite adequate treatment based on the undetectable viral load or absence of surface protein, there has been much research demonstrating persistent risk for HBV-associated HCC. The aim of this paper is to review the related factors, pathophysiology, and evidence for why this risk exists. Further clarification of the relationship and risk factors for HBV-related HCC is necessary for appropriate screening and the eventual development of a cure.

**Abstract:**

Chronic hepatitis B virus (HBV) infection is the largest global cause of hepatocellular carcinoma (HCC). Current HBV treatment options include pegylated interferon-alpha and nucleos(t)ide analogues (NAs), which have been shown to be effective in reducing HBV DNA levels to become undetectable. However, the literature has shown that some patients have persistent risk of developing HCC. The mechanism in which this occurs has not been fully elucidated. However, it has been discovered that HBV’s covalently closed circular DNA (cccDNA) integrates into the critical HCC driver genes in hepatocytes upon initial infection; additionally, these are not targets of current NA therapies. Some studies suggest that HBV undergoes compartmentalization in peripheral blood mononuclear cells that serve as a sanctuary for replication during antiviral therapy. The aim of this review is to expand on how patients with HBV may develop HCC despite years of HBV viral suppression and carry worse prognosis than treatment-naive HBV patients who develop HCC. Furthermore, HCC recurrence after initial surgical or locoregional treatment in this setting may cause carcinogenic cells to behave more aggressively during treatment. Curative novel therapies which target the life cycle of HBV, modulate host immune response, and inhibit HBV RNA translation are being investigated.

## 1. Introduction

Hepatitis B virus (HBV) is a global public health problem with an estimated 350 million chronic hepatitis B carriers, causing 820,000 deaths in 2019 alone [1,2,3]. HBV is endemic in many parts of the world, including Southeast Asia, China, and Africa [4]. It belongs to a family of DNA viruses called hepadnaviruses, which is composed of at least ten genotypes, A–J [2,5,6]. The virion is an enveloped nucleocapsid that delivers an incomplete circular DNA genome into the host cell, initiating viral replication [7]. HBV is a dynamic, hepatotropic virus and its infection has a wide spectrum of clinical manifestations [7]. Fifteen–forty percent of HBV-infected patients develop cirrhosis, liver failure, or hepatocellular carcinoma (HCC) [1,8]. In fact, HBV is the most common hepatocarcinogen, being accountable for 25% of HCC cases in developed countries and 60% in developing countries [9,10,11,12]. There is a limited understanding of the pathogenesis, prognosis, and treatment strategies for HBV-associated HCC, which we highlight in this review paper.

## 2. HBV Pathophysiology and Hepatocarcinogenesis

HBV is transmitted by percutaneous inoculation or transmission of infectious bodily fluids. In high-prevalence areas, mother-to-child transmission is the predominant mode of transmission, while unprotected sex and injection drug use are the common modes of transmission in low-prevalence areas [13,14]. The incubation period of HBV is between 30 and 180 days. During infection, complete and incomplete viral particles are released into the serum of the host, facilitating viral replication [15].

HBV is a non-cytopathic virus, and the liver damage associated with HBV is caused by the host immune response [16,17]. During acute infection, host immune cells, most prominently CD8 T cells, kill infected cells, inducing hepatic inflammation [4,18]. Around 70% of patients with acute HBV have subclinical or anicteric hepatitis, while 30% have icteric hepatitis [19]. The clearance of HBV is mediated by the adaptive immune system and HBV utilizes multiple strategies to evade this line of defense [19,20,21]. The hypo-responsiveness of HBV-specific T cells may also contribute to persistent HBV infection [21]. While recovery commonly occurs in immunocompetent individuals, a small proportion of those infected can progress to chronic HBV infection, which is defined as the presence of HBsAg for greater than six months [22].

There are four phases of chronic hepatitis B (CHB) infection: immune tolerance, immune clearance, immune control, and immune escape/reactivation [23]. In the immune tolerance phase, ALT levels are still low, viral DNA levels are high (usually at least 2,000,000 IU/mL), and there is minimal or no inflammation on liver biopsy [4,18,23,24,25,26]. This phase may last for a few years to around 30 years [27,28]. In the immune clearance phase, hepatitis B e antigen (HBeAg) is positive, there is intermittent or persistent elevation of ALT levels, elevated HBV DNA levels (at least 2000 IU/mL), and some degree of inflammation or fibrosis on liver biopsy [23,28]. In this phase, HBV-specific CD8 T cells directly attack infected hepatocytes and recruit other immune cells to the liver, which further exacerbate hepatic injury [4,17,18,26,29]. During immune clearance, patients may present with flares, which—while often being asymptomatic—may be characterized by signs of acute hepatitis. During flares, ALT levels may be elevated to greater than five times the upper limit of normal [23,30]. The end of the immune clearance phase and beginning of immune control is marked by seroconversion, which is the loss of HBeAg and development of antibodies to hepatitis B e antigen (HBeAb). Intriguingly, the duration of the immune clearance phase has a critical association with the development of complications; those who seroconvert after the age of 40 have a significantly higher risk of cirrhosis, HCC, and CHB compared to those who seroconverted before the age of 30 [23,31]. In addition, HBeAg positivity is a known risk factor for HCC [26,32]. The immune control phase is characterized by lower levels of HBV DNA (usually <2000 IU/mL) and ALT, although HBsAg remains [4,18,23]. Some patients, even after seroconversion, may continue to have moderate levels of viral replication with associated abnormal ALT, leading to eventual reactivation of the immune active phase; this phenomenon is known as immune escape [23,28,33,34]. Resolution of infection is indicated by disappearance of HBsAg [4,18].

In CHB, there are crucial changes in immune cell activity and function involving both the innate and adaptive immune systems that lead to hepatic inflammation and hepatocyte killing [17]. Chronic infection may progress to liver fibrosis, cirrhosis, and HCC [17,35]. HBV-mediated carcinogenesis is a complex process that involves viral DNA integration into the host genome, ultimately leading to viral manipulation of cell-signaling and proliferation. This leads to a cascade of events which converts normal hepatocytes into malignant cells [36,37,38,39]. Oxidative stress associated with viral hepatitis changes the cellular environment in such a way that promotes carcinogenesis. Overproduction of free radicals and reactive oxidative species leads to the upregulation of inflammatory pathways that result in hepatocyte release of cytokines and chemokines that recruit neutrophils, monocytes, and lymphocytes [18,37,40,41]. As inflammation persists, immune cells, including macrophages and myeloid-derived suppressor cells, become dysfunctional, further amplifying the pro-inflammatory environment. The chronic inflammatory state leads to compensatory hepatocyte proliferation, which leads to accumulation of mutations that promote cell growth and proliferation which predisposes the host to developing HCC (Figure 1) [40,42].

Moreover, it was found that the incidence of HCC is fivefold higher among HBV-infected patients with cirrhosis compared to asymptomatic carriers, indicating that cirrhosis may be a pre-malignant condition [38]. Indeed, fibrosis of the liver disrupts the normal architecture which leads to modification of cell–cell interactions and ultimately, loss of regulation over cell proliferation [38].

Furthermore, it is important to note that numerous factors including host characteristics, HBV genotype, viral mutations, viral load, and HBsAg levels all influence the clinical manifestations of HBV infection [6]. Research has shown that there are differences between the various genotypes of HBV in tendency of chronicity, primary mode of transmission, timing of seroconversion, timing of HBsAg clearance, and clinical outcomes [6]. For example, multiple studies have shown that genotype C tends to cause more severe liver disease, including cirrhosis and HCC, compared to other genotypes [6,43,44]. Genotype C also has higher serum HBV levels and was shown to cause DNA mutations more frequently than Genotype B [6,45]. Further research investigating the utility of routine HBV genotyping is needed before implementation into clinical practice. 

## 3. HCC Risk Factors and Surveillance

HCC is one of the major malignant diseases in the world today and ranks fifth in overall frequency. Its incidence is high in Eastern Asia and sub-Saharan Africa and is increasing in many parts of the Western world [46]. Among cancers, the annual mortality from HCC is relatively high because of its rapid progression and poor prognosis. Unfortunately, the diagnosis is typically made later in its course when therapeutic interventions are generally ineffective. Thus, the focus has been on early screening and treatment of known causes, including chronic hepatitis B which is the most frequent underlying cause of HCC. There are several major risk factors for development of HCC in chronic HBV infection. While having a first-degree relative with HCC, metabolic syndrome, type 2 diabetes, and central obesity are all host factors that have been linked to HCC development in HBV-infected patients or carriers, it appears that liver cirrhosis has been consistently identified to be the most significant risk factor for development of HCC during nucleos(t)ide therapy [47,48,49,50,51]. Notably, there is increasing evidence that non-alcoholic fatty liver disease (NAFLD) is becoming one of the largest causes of HCC, particularly in industrialized countries [52]. In these patients, HCC can develop even in the absence of cirrhosis. As there are no specific pharmacological therapies for NAFLD, adoption of healthy lifestyle changes including weight loss and regular aerobic physical activity remain the mainstay treatment. Interestingly, bariatric surgery is known to provide durable weight loss and has been identified to be associated with a decreased risk of HCC. A meta-analysis involving nearly 20 million patients showed that bariatric surgery has a protective effect on risk of HCC occurrence and incidence compared to those subjects who did not undergo the procedure [53]. Studies indicate that the presence of male gender, increasing age, higher HBV DNA level, and core promoter mutations are also independently associated with HCC risk [54,55]. It is postulated that androgens may play a role in the observed difference in incidence of HCC based on gender [56]. Among patients with chronic hepatitis B infection treated with nucleos(t)ide therapy, those with hepatitis D coinfection have been shown to have a nearly sixfold risk of HCC development compared to patients without hepatitis D coinfection [57]. While less investigated, concurrent HCV or HIV infection has also been identified to have an association with increased incidence of HCC [58,59]. The risk factors for hepatocarcinogenesis can be seen in Figure 2.

Surveillance with ultrasound imaging and measurement of alpha-fetoprotein levels has been directed towards these populations to help with early detection of HCC. Guidelines suggest surveillance for HCC in high-risk groups including patients with cirrhosis, noncirrhotic patients with HBV and any of the following characteristics: active hepatitis, family history of HCC, Africans and African Americans, Asian males over 40 years of age, and Asian females over 50 years of age [60,61,62,63]. Other populations that undergo surveillance include patients with chronic hepatitis C virus and advanced liver fibrosis in the absence of cirrhosis, although this is not a consistent recommendation across all societies and the cost effectiveness has not been verified [61]. 

Currently a knowledge gap remains regarding host factors that contribute to the vagaries of HBV infection outcomes. A group of researchers attempted to explore this by investigating the diverse manifestations of CHB in three families that were observed over decades. Block et al. showed how only one case of HBV-related HCC occurred within every family cluster despite each having the same virus given perinatal transmission from mother to offspring [64]. Furthermore, one of the families had monozygotic twins in which only one sibling developed HBV-related HCC, while the other remained a chronic HBV carrier. The same finding is presenting in a case series by Noverati et al., which presented four family clusters in which patients had very variable courses, some with indolent chronic HBV infection, some requiring treatment, and others who developed HCC or cirrhosis [65]. It is postulated that inheritable immuno-genetic alleles that affect CHB differ from those that influence HBV-related HCC development, which may explain the discrepancy in manifestations in this case series [66]. The study was limited by lack of host and viral genomic analysis, as previous studies have shown certain host polymorphisms and HBV mutations are associated with HBV-related HCC. Nonetheless, this highlights that there are genetic and non-genetic host factors that play a role in development of HCC.

Exogenous, non-genetic factors such as chronic stress have been implicated in the incidence and outcomes of HCC, which may also have contributed to the differing presentations in the case outlined above [18]. One of the first published studies highlighting the significance of stress was by Russ et al., in which the group found in a large UK meta-analysis that those with higher scores on a general health questionnaire measuring psychological distress had a higher mortality from liver disease [67]. Joung et al. outlined how stress can increase inflammatory, oxidative reactions including hypoxia–reoxygenation, overactivation of Kupffer cells, influx of gut-derived lipopolysaccharide and norepinephrine, and overproduction of stress hormones in the sympathetic drive, which cause hepatocellular damage and promote mutagenesis [68]. Studies have shown how the tumor milieu of existing HCC undergoes changes that make it immunosuppressive. Specifically, He et al. discovered how chronic stress transitions cytokines to those that are T helper 2 cell-mediated, a tumor microenvironment that carries a poorer prognosis [69]. Similar studies show how this is exacerbated by the presence of T-regulatory cells that suppress pro-inflammatory cytokines [70,71].

## 4. Incidence of HCC in Response to Treatment of HBV

Chronic inflammation and necrosis are key factors which predispose patients to developing HCC. In HBV infection, viral load generally correlates with the severity of disease. NA therapies aimed at decreasing viral load include lamivudine, entecavir, and tenofovir, the latter two of which are standard of care and have provided a breakthrough in the treatment of HBV infection as they have demonstrated a reduction in incidence of HCC [72,73,74]. Results of prior systematic reviews demonstrate that the risk of HCC is significantly lower in CHB patients receiving oral antiviral therapy. Of the first data to demonstrate efficacy of anti-HBV therapy in HCC risk reduction, some came from a randomized controlled trial by Liaw et al. which compared lamivudine vs. placebo in treatment-naive patients with cirrhosis or advanced fibrosis and active liver disease. Lamivudine treatment for 3 years reduced the risk of HCC by 51% compared to the placebo [74]. According to Papatheodoridis et al., across three studies including untreated controls, HCC developed in 2.8% of treated and in 6.4% of untreated CHB patients (*p* = 0.003). A similar study by Singal et al. pooled data from six other studies and showed that the incidence of HCC among lamivudine-treated patients was significantly lower than that of untreated patients (3.4% vs. 9.6%, *p* < 0.0001) [75]. These data also demonstrated that HCC occurred more frequently among CHB patients receiving therapy who were older, had cirrhosis, or had detectable HBV-DNA at the end of treatment. 

Over time, it was discovered that several patients on lamivudine therapy did not attain virological response in the setting of resistance. This did not come without consequence. Alarmingly, Papatheodoridis et al. demonstrated that the rate of HCC was significantly higher in lamivudine resistance than in nucleos(t)ide-naive patients (7.1% vs. 3.8%, *p* = 0.001), even after correcting for those with cirrhosis (18% vs. 11%, *p* = 0.015). It is believed that mutations including changes at position 181 of the reverse transcriptase domain of HBV polymerase have oncologic potential. This led to the advent of newer nucleos(t)ide analog therapies such as tenofovir and entecavir, which proved to be effective. Kim et al. observed that the incidence of HCC in patients with CHB and cirrhosis was decreased compared to the predicted incidence for untreated CHB patients after treatment with tenofovir disoproxil fumerate (TDF) for 3 years [73]. Lim et al. elaborated on this finding and showed that the difference in observed versus predicted HCC risk is more pronounced with tenofovir alafenamide (TAF) than TDF treatment [76]. Hosaka et al. demonstrated that HCC suppression effect was greater in entecavir-treated than non-rescued lamivudine-treated cirrhosis patients when compared to the control group [72]. This was supported by Papatheodoridis et al. in a prospective cohort study of HBeAg-negative CHB patients which showed that entecavir compared with lamivudine group patients had lower HCC incidence [77]. In a retrospective cohort of 451 newly diagnosed HBV-related HCC patients, lamivudine use, compared to entecavir, was associated with hepatic decompensation and recurrence of HCC after curative therapy [78]. In a study of entecavir-treated patients, the cumulative incidences of HCC were lower with achievement of a virologic response in both non-cirrhotic and cirrhotic patients [79]. In another study which included cirrhotic CHB patients treated with entecavir, those with undetectable HBV DNA levels were associated with a lower probability of developing HCC [80]. A separate study sought to identify the long-term clinical impact of low-level viremia (<2000 IU/mL) in patients on entecavir monotherapy. Kim et al. observed that, among patients with cirrhosis, those with low-level viremia exhibited a significantly higher HCC risk than those with maintained virological response [81]. These findings support the current recommendation for indefinite therapy in chronic HBV patients, and that inability to achieve undetectable viral load can be consequential. 

## 5. Persistent Risk of HCC in HBV Infection

Maintenance of viral suppression is necessary to reduce HCC risk, as demonstrated above. Unfortunately, recent evidence shows that there remains a persistent risk of HCC despite successful HBV therapy resulting in viral suppression [46,82,83,84,85,86]. While spontaneous or treatment-induced seroclearance of HBsAg is also associated with lower risk of liver-related complications, there remains persistent risk of HCC development in these patients as well. Ahn et al. showed a significant reduction in necroinflammation on liver biopsy of patients before and after seroclearance. However, all the patients demonstrated the presence of HBV DNA without a change in the fibrosis score [87]. Additionally, Gounder et al. showed that seroclearance was not associated with reduced HCC risk compared to a matched control group [88]. The proposed mechanisms in which HCC develops despite adequate treatment are described here.

It has been discovered that HBV’s covalently closed circular DNA (cccDNA) integrates into the hepatocytes upon initial infection and can be detected even after antiviral treatment. This integration may occur at critical HCC driver genes which may give rise to malignant HBV-infected hepatocytes. Li et al. found that the common location of integration included the *TERT*, *CCNEI1*, and *MLL4* genes, and that virus–host chimera DNA circulate in 97.7% of CHB patients with HCC [89,90]. Interestingly, the group demonstrated how virus–host chimera DNA could be used as a biomarker for detecting tumor load in this patient population, and how it may aid in monitoring recurrence after tumor resection [89]. In the first detailed molecular evaluation completed by Coffin et al., despite undetectable HBV DNA in plasma via clinical assays, patients treated with oral anti-HBV therapy undergoing liver transplant were found to have low levels of HBV genome in the liver, circulating lymphoid cells, as well as plasma using ultrasensitive PCR/nucleic acid hybridization assay. Their data promote the idea of HBV compartmentalization, where despite adequate suppression, the liver supports replication and extrahepatic HBV can be detected, particularly in peripheral blood mononuclear cells (PBMCs) that may serve as a sanctuary for wild-type viruses during antiviral therapy [91].

Other possible mechanisms that have been postulated include progression of cirrhosis, necroinflammation caused by HBV infection, and the oncogenic effect of HBV genome integration into the hepatocyte chromosomes. HBV DNA may remain detectable in the serum or liver cells, which is referred to as occult HBV infection [92]. Wong et al. found that 69% of 90 HBsAg-negative patients with HCC had occult infection and that of these patients, 47% had detectable cccDNA in hepatocytes and 53% had integration of HBV DNA near hepatic oncogenes [93]. In fact, the HBV genome has been detected in tumor tissue of HBsAg negative patients with HCC in a prevalence ranging from 30% to 80% [94,95]. Therefore, these patients continue to undergo surveillance per the AASLD guidelines [96]. Michalak et al. showed that traces of the HBV genome persist for years following recovery from hepatitis B (characterized by seronegativity) and that they can lead to HCC development in a woodchuck model [97].

HBV infection has an inhibitory effect on the innate and adaptive immune cells which may aggravate chronic inflammation and lead to carcinogenesis. These effects are thought to persist despite antiviral treatment as the microenvironment is altered upon initial infection. Data suggest that this immune-tolerant phase of CHB infection is when hepatocarcinogenesis begins, with integration of HBV DNA into the genome of hepatocytes [98]. A retrospective cohort study by Kim et al. demonstrated how untreated patients in an immune-tolerant phase had 2–3 times higher risk of HCC, liver transplantation, or death compared with patients in immune clearance phase who were initiated on NA therapy [99]. The immune-tolerant phase was defined as HBeAg-positive, HBV DNA level greater than or equal to 20,000 IU/mL and normal ALT for one year. It has been shown that the virus may decrease the expression of STAT3 in NK cells, ultimately inhibiting their ability to clear HBV [100]. Studies demonstrate that HBV may promote polarization of cells from M1 to M2 and inhibit secretion of antiviral cytokines by macrophages [101]. The idea that there are frequent integration events and detectable HBV viral load during the immune-tolerant phase suggests that there may be a role of early introduction of NA therapy to decrease risk of HCC and fibrosis progression [102,103]. This would ultimately mean initiation of therapy in young adults with immune-tolerant profile. However, the low rates of response, the costs of long-term treatment, and the lack of established benefits in prevention of HCC are obstacles to be considered [104,105]. Further research on the immune-tolerant population is necessary to fully understand this dynamic. Immune tolerance to HBV infection plays a key role in the acceleration of progression of HBV to HCC.

Most reported cases of HBV-HCC in the literature are within the first five years of NA therapy, but data on long-term risk of HCC while on treatment are lacking. The Liver Disease Prevention Center, Division of Gastroenterology and Hepatology at Thomas Jefferson University Hospital, has observed several HBV positive patients develop HCC even after 10 or more years of successful viral suppression [106]. Specifically, in our observational cohort, 17 patients developed HCC despite having noted undetectable HBV DNA for up to 12 years and on treatment for up to 19 years (Table 1) [107]. Currently, this is among the longest known follow-up studies on the development of HCC in patients on HBV treatment.

Our center has observed six patients who developed new or recurrent HCC after undergoing curative therapy for HCC and achieving adequate HBV suppression for 5–11 years on antivirals [106]. Unfortunately, recent studies have shown that the patients who develop HCC after years of viral suppression, in fact, carry worse prognoses than those who develop HCC without prior treatment of HBV [108,109]. A case series of 26 patients, 13 antiviral-naive and 13 antiviral-treated, assessed clinical outcomes after HCC diagnosis. Garrido et al. showed that after the first HBV-HCC event, death was observed more frequently among the antiviral-treated cohort at 46.2% compared to 15.4% in the antiviral-naive cohort [108]. Li et al. compared patients who received NA therapy after hepatectomy for HCC and those who did not receive treatment and discovered that there was no difference in the recurrence rate of HCC [110]. The suggested mechanism by which this occurs is that halting HBV replication at the reverse transcription phase for a prolonged time may prime the infected hepatocytes to increasingly turnover HBV DNA integration, which can cause insertional mutagenesis and genomic instability, in the setting of HBV compartmentalization and immune-tolerant environment, as previously described (Figure 3) [111]. This additionally results in the accumulation of incompletely transcribed mRNAs which may lead to further integration.

It is believed that HCC inoculation in this setting of disorganized products and instability may also cause cancerous cells to be primed to behave more aggressively. Chowdhury et al. described two cases of HBV-treatment-naive HCC where despite radiofrequency tumor ablation (RFA) and subsequent undetectable HBV viral load on NA therapy, the patients developed recurrent, more aggressive cancer [109]. The first case described an elderly female who underwent trans-arterial embolization (TACE) of a 5 × 2.1 cm mass suggestive of HCC. She started NA therapy and became HBV-DNA-negative. However, she developed recurrence 5 years later and required repeated treatment of TACE and RFA. Ten years after the recurrence, MRI demonstrated another 5 mm LI-RADS 4 lesion in segment 2 which was monitored with evidence of regression. Unfortunately, this tumor had grown to 1.2 cm two years later, which marked two decades after the development of her initial HCC (Figure 4) [109]. The second case described a male who was found to have cirrhosis with 1.0 cm HCC at age 68. He underwent NA therapy with undetectable HBV DNA and RFA; however, at age 79, he was found to have two new lesions on an abdominal MRI (3 × 2.8 cm and 2 × 1.7 cm) in segment 5, adjacent to the gallbladder, and segment 7/8, respectively. He received a cholecystectomy and tumor ablation but four years later was found to have a new 4.8 cm lesion in segment 4 on MRI (Figure 5) [109]. This illustrates the complexity of the relationship between HBV and HCC, as there may be accelerated progression and recurrence of tumor in patients after becoming treatment-experienced. Notably, this is a preliminary observation that requires larger studies to characterize, as random integration of HBV DNA and anatomical location of initial HCC lesion also play key roles in the development of further HCC events. Nonetheless, these cases place emphasis on the value of adhering to scheduled screening and monitoring of HCC recurrence years after initial diagnosis.

Numerous studies have been conducted investigating the role of HBV infection in resistance to sorafenib therapy in patients with HBV-associated HCC. Newer systemic chemotherapies have been approved for the treatment of HCC, such as atezolizumab/bevacizumab, durvalumab/tremelimumab, and lenvatinib; however, for more than 10 years, sorafenib has been considered to be the standard of care for patients with advanced unresectable HCC. Sorafenib inhibits activity of many tyrosine kinases involved in tumor angiogenesis and progression including vascular endothelial growth factor receptor, platelet-derived growth factor receptor, Flt3 and c-Kit, and many Raf kinases involved in the MAPK/ERK pathway [112]. These pathways ultimately lead to proliferation, migration, survival, and metastasis of tumor cells. Liu et al. has shown that there is overexpression of variant 1 (tv1) of proliferating cell nuclear antigen clamp-associated factor (PCLAF) particularly in HBV-related HCC than in non-virus-related HCC [113]. In their biochemical studies, HBV promoted splicing of PCLAF tv1 through downregulating serine/arginine-rich splicing factor 2 (SRSF2). Through suppression of ferroptosis and regulation of abnormal splicing of the SRSF2/PCLAF tv1 axis, HBV was noted to induce sorafenib resistance. In a separate study by Liu et al., immunohistochemistry staining of HBC-associated HCC tissue demonstrated overexpression of WNT7B compared to nontumor liver tissues [114]. This signaling molecule along with its receptor frizzled-4 (FZD4) were noted to be upregulated in response to large hepatitis B surface antigens (L-HBs), which increased WNT signaling in HCC cells, leading to proliferation and metastasis. By this mechanism, L-HBs suppressed sorafenib-induced mitophagy, diminishing the therapeutic effect of chemotherapy. Zhang et al. demonstrated in mouse models that cellular inhibitor of apoptosis protein 2 (cIAP2) plays a key role in promotion of cell survival of carcinogenic hepatocytes in HBV-associated HCC [115]. Cell lines rich in cIAP2 had reduced caspase levels and increased cell viability when treated with sorafenib. However, when these cells were treated with lamivudine, cIAP2 expression decreased and sorafenib sensitivity was partially restored.

Other studies have investigated potential target strategies to overcome this resistance including regulatory T cells (Treg). CCR4+ Tregs are the predominant type of Tregs that are recruited to HBV-associated HCC and directly correlate with HBV titers, exhibiting increased immunosuppressive cytokines such as IL-10 and IL-35 [116]. It was shown in a mouse model study by Gao et al. that treatment with CCR4 antagonist could block Treg accumulation to overcome sorafenib resistance, and potentially serve as a therapeutic target in HCC immunotherapy [116]. This research group expanded on this finding by suggesting that blocking macrophage-derived chemokine (CCL22), which controls migration of Tregs via its receptor CCR4, may also be an immunotherapeutic strategy for HCC through a similar mechanism [117]. Notably, there is an increased risk of severe autoimmunity without adequate regulation by Tregs. Other mechanisms to sensitize HBV-associated HCC to sorafenib-induced apoptosis include modulation of miRNA such as miR-193b and miR-3677-3p, in addition to inhibiting pro-oncogenic Mitogen-Activated Protein Kinase 14 (pMAPK14) [118,119,120].

NAs have been critical in decreasing HBV-associated mortality worldwide, However, we demonstrate in this review that they do not eradicate the risk of HCC development. There is persistent risk of hepatocarcinogenesis despite adequate treatment marked by undetectable HBV DNA. Thus, there remains a public health threat as currently there is no cure for HBV infection. Achievement of a cure would require therapies which eliminate cccDNA and integrated HBV DNA as well as help restore the impaired innate and adaptive immune responses against HBV.

## 6. The Road to HBV Cure

The AASLD defines two different types of “cures” of HBV. First, it defines “immunological cure” as HBsAg loss and sustained HBV DNA suppression. Second, it defines “virological cure” as eradication of the virus, including cccDNA, which is not achievable with current therapies. 

The next generation of drugs in development to treat hepatitis B can be divided into the following classes: drugs targeting the life cycle of the hepatitis B virus; drugs aiming to modulate the host’s immune response to CHB infection; monoclonal antibodies aiming to directly inhibit translation of HBV RNA; and therapeutic vaccines (Figure 6) [121]. A detailed discussion of the progress of these drugs’ development is beyond the scope of this paper; for further reading, the excellent review by Yardeni et al. is highly recommended [122].

Drugs targeting the life cycle of the hepatitis B virus can be classified into entry inhibitors, capsid assembly modulators, post-transcriptional control inhibitors, and hepatitis B surface antigen release inhibitors.

Novel drugs aiming to modulate the host’s immune response to the HBV are comprised of innate immune activators—specifically, Toll-like receptor (TLR) agonists, RIG-1 agonists, and checkpoint inhibitors. 

Monoclonal antibodies designed to inhibit the life cycle of HBV have been investigated. One such neutralizing monoclonal antibody, VIR-3434, enhances HBsAg uptake by dendritic cells to promote its presentation to naive T cells thereby stimulating an HBV-specific immune response [123].

Finally, there has been interest in using therapeutic vaccines in effort to stimulate the host’s immune response against CHB infection. However, early studies in humans showed only modest declines in HBsAg [124]. With that said, several new therapeutic vaccines are in development which may be tested in combination with drugs of other mechanisms of action, including siRNAs [125]. However, these therapies have not yet been tested in humans. 

In summary, the emerging landscape for novel CHB treatments is composed of vast and varied mechanisms of action, but further clinical testing is still required to establish superiority in efficacy compared to the current standard of care.

## 7. Conclusions and Future Directions

HBV is a global health threat worldwide as infection can lead to HCC. Available treatment options do not fully eradicate the risk of hepatocarcinogenesis as initial integration of cccDNA into the host genome is not prevented. Continued transcription creates a milieu of genomic instability which can make recurrence of HCC even more aggressive. The phenomenon of worse prognosis in treatment-experienced HCC versus treatment-naive HCC is observed. While there remains a persistent risk of HBV-associated HCC despite viral suppression, recent studies show promising novel approaches to HBV cure.

## Figures and Tables

**Figure 1 cancers-16-00777-f001:**
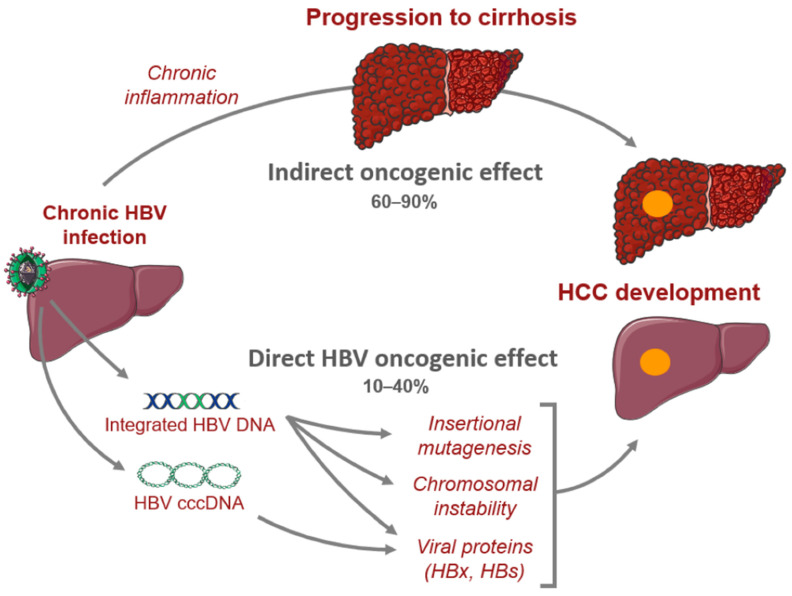
Mechanisms of hepatocarcinogenesis from chronic hepatitis B infection. Sourced from Péneau et al. [42].

**Figure 2 cancers-16-00777-f002:**
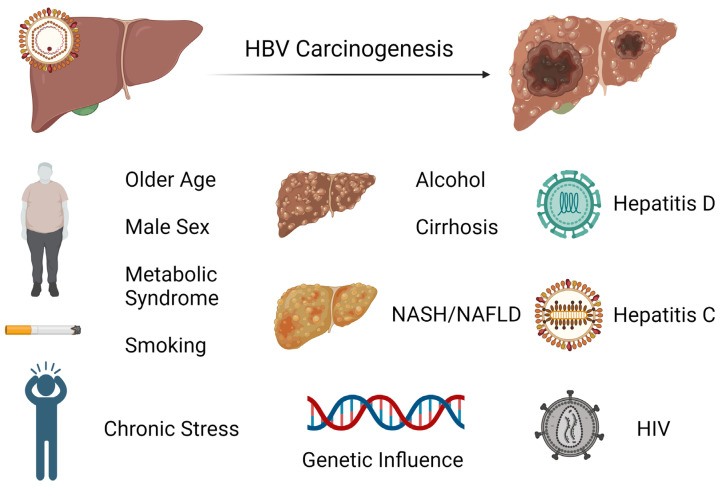
Exogenous and endogenous risk factors for hepatocarcinogenesis from chronic hepatitis B infection. Created with BioRender.com (accessed on 21 January 2024).

**Figure 3 cancers-16-00777-f003:**
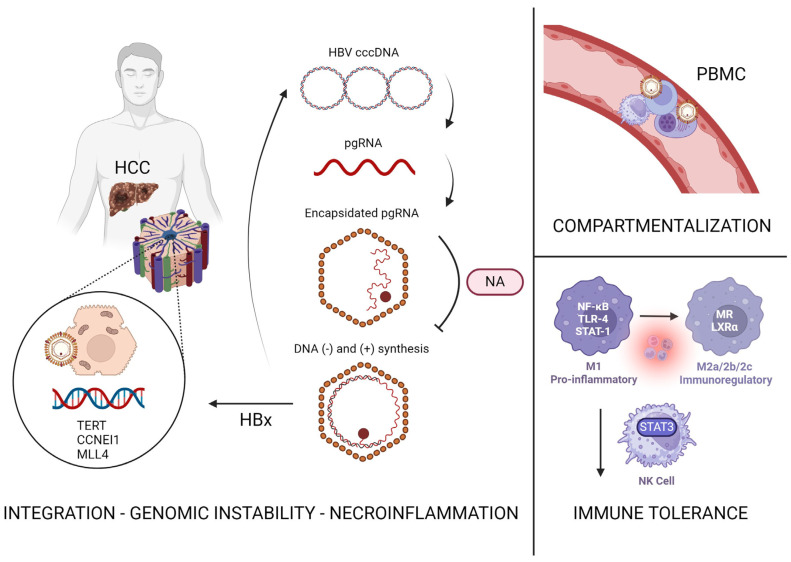
Proposed mechanisms behind persistent risk of HBV-associated HCC. cccDNA—covalently closed circular DNA; pgRNA—pre-genomic RNA; NA—nucleos(t)ide analog; HBx—HBV virus X protein; PBMC—peripheral blood mononuclear cells; NK—natural killer. Created with BioRender.com (accessed 24 January 2024).

**Figure 4 cancers-16-00777-f004:**
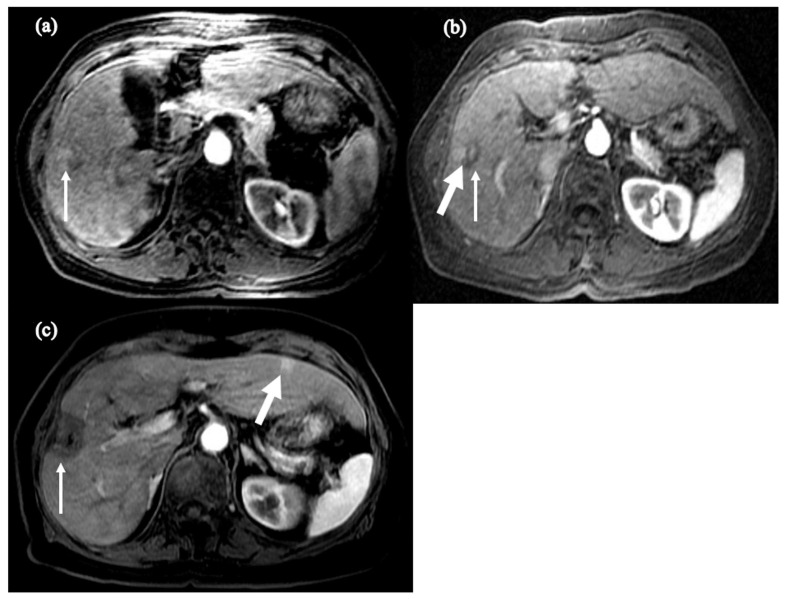
Axial fat-suppressed, T1-weighted postcontrast images of (**a**) the initial mass concerning for HCC (arrow), (**b**) post-TACE lesion (thin arrow) with adjacent focal hyperenhancement (thick arrow), and (**c**) post-TACE and RFA lesion in segment 5 (thin arrow) with new hyper-enhancing lesion in segment 3 (thick arrow). Sourced from Chowdhury et al. [109].

**Figure 5 cancers-16-00777-f005:**
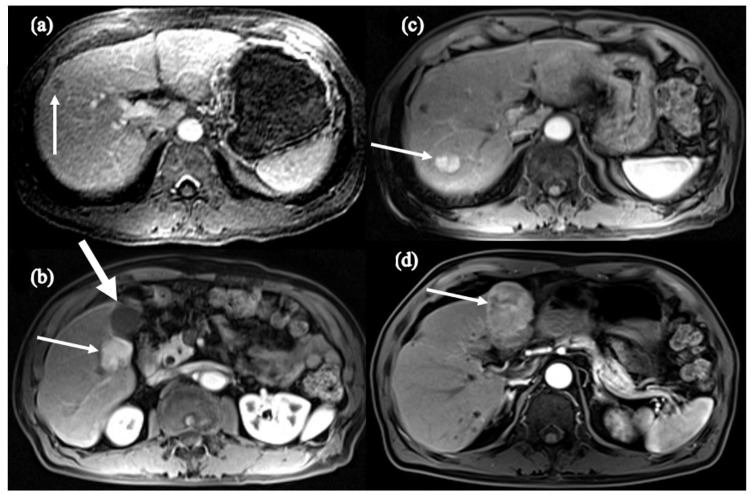
Axial fat-suppressed, T1-weighted postcontrast images of (**a**) the initial segment 8 mass concerning for HCC (arrow), (**b**) post-RFA and NA therapy 3 cm segment 5 hyper-enhancing HCC (thin arrow) adjacent to gallbladder (thick arrow), (**c**) second hyper-enhancing segment 7 HCC (arrow), and (**d**) post-cholecystectomy and tumor ablation large segment 4 HCC (arrow). Sourced from Chowdhury et al. [109].

**Figure 6 cancers-16-00777-f006:**
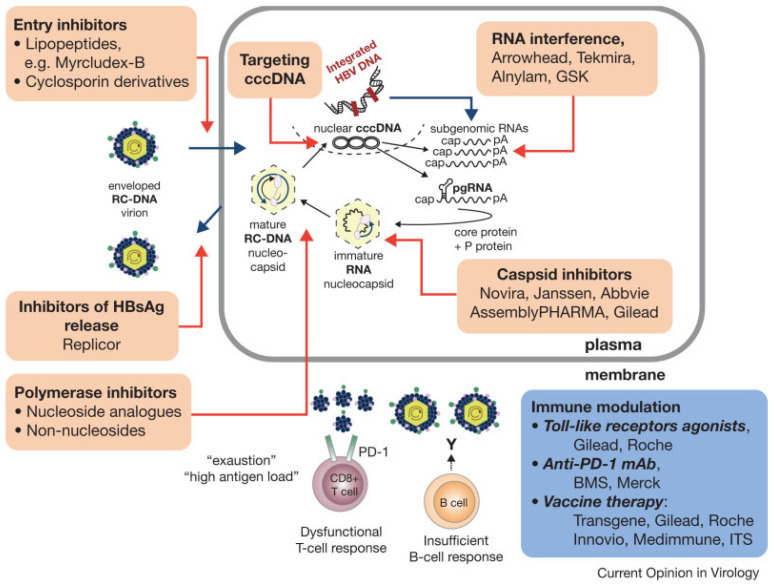
Therapeutic drug targets of hepatitis B virus replication. Sourced from Levrero et al. [121]. Used with permission from original copyright holder.

**Table 1 cancers-16-00777-t001:** Development of HCC in patients with cirrhosis on long-term antiviral therapy. Sourced from Boortalary et al. [107].

Pt	Date StartTx	Change in Child Class on Tx	Date HCC Dx	Yrs on Anti-HBV Tx at HCC Dx	Yrs with HBV DNA (-) *	Age (Yr) at HCC Dx	Tumor Size at Dx	HBVDNA at HCC Dx	Anti-HBV Tx	Status
1	April 1998	B → A	July 2007	9	3	53	1.1 Junction	UD	LAM + TDF	Alive
2	January 1998	B → A	March 2008	10	8	68	2.8 Rt	UD	LAM + TDF	Dead
3	May 1998	A →A	February 2008	10	7	76	1.8 Lt	UD	LAM + TDF	Alive
4	July 2001	B → B	September 2010	9	4	54	2.8 Rt	UD	LAM + TDF	Dead
5	August 2004	B → B	November 2010	16	4	53	3.9 Rt	UD	LAM + TDF	Alive
6	July 2001	B → B	January 2011	10	5	55	2.8 Rt	UD	LAM + TDF	Dead
7	February 2004	A → A	June 2013	9	8	57	2.5 Lt med	UD	TDF	Dead
8	February 1996	A → A	July 2013	17	10	73	1.6 Rt	UD	TDF	Dead
9	August 1997	A → A	June 2014	17	6	54	2.2 Lt lat	UD	ETV	Alive
10	March 2004	B → B	June 2013	9	7	57	2.5 Lt	UD	TDF	Dead
11	July 2001	A → A	June 2014	13	7	54	2.2 Lt	UD	TDF	Alive
12	May 1996	A → A	October 2014	18	10	74	3.4 Rt	UD	LAM + TDF	Dead
13	February 2000	A → A	October 2014	14	12	62	3.8 Rt	UD	ETV + TDF	Alive
14	February 2000	A → A	April 2015	15	12	62	3.4 Rt	UD	TDF	Alive
15	February 2000	B → A	May 2015	15	12	65	3.8 Rt	UD	TDF	Alive
16	December 1998	A → A	August 2017	19	8	64	2.0 Rt	UD	LAM + TDF	Alive
17	February 2008	A → A	June 2019	11	10	57	2.2 Rt	UD	ETV + TDF	Alive

Dx: Diagnosis, ETV: Entecavir, LAM: Lamivudine, Lt: Left, Pt: Patient, Rt: Right, TDF: Tenofovir Disoproxil Fumarate, Tx: Treatment, UD: Undetectable, * Years with HBV DNA (-) at diagnosis of HCC.

## Data Availability

No new data were created or analyzed in this study. Data sharing does not apply to this article.

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
