# Peer review of "Review of Related Factors for Persistent Risk of Hepatitis B Virus-Associated Hepatocellular Carcinoma"

_cancers, 2024, doi:10.3390/cancers16040777_

Round 1

Reviewer 1 Report

Comments and Suggestions for Authors

line 65: chronic HBV infection ? --> Chronic hepatitis B (CHB) (HBV as your line 38, is a virus)

line 127: the annual mortality from HCC  is close to its incidence because of its rapid progression and poor prognosis : could you cite where this conclusion comes from and which article, it seems is not acceptable to the doctors who work on HBV?

line 128: Additionally,  the therapeutic interventions available at the time of diagnosis are generally ineffective = it only happened in late presentation.

line 140: no mention of HDV infection (https://www.who.int/news-room/fact-sheets/detail/hepatitis-d) which co-infection will make anti-HBV medication fail

line 187: "the latter two (or TAF)" are now standard of care, no more about lamivudine.

line 270: Immune tolerance to HBV infection plays a key role in the acceleration of progression of HBV to HCC =  please describe it more, cited from Kali Zhou and Norah Terrault ?. Should we begin NA treatment in this group of patients?

line 281: 41% died, no HCC staging, 24% BCLC A0 or? no mention of tumor numbers  

line 291: naive cohort [99].--> 99 should be 97

line 303: the first and line 309 the second: this section comes from reference 98 but the reader did not see the pictures on the original paper, these two in the original article are cases 4th and 5th

line 304: The size should be the larger diameter in the front: 5 x 2.1 cm (as you describe on line 312), The down-size procedure should be TACE then RFA

line 309: 28 years ? only 20 years

line 317: I think cases 4 and 5 might be too old to have frequent examinations, you should emphasize the scheduled screening but not the idea of prolonged oral NA taking will be harmful, that will make the patients confused and discuss stopping NA which the latter is the purpose of professor Liaw (reference 19). The random integration made the difference in the aggressiveness of HCC and the location of HCC near the vessels makes the distribution of intrahepatic or distal metastases, not only your opinion of long antiviral therapy.

line 380: I do not know what this DOI means, just a guidance to a Book Review?

Reviewer 2 Report

Comments and Suggestions for Authors

Chronic hepatitis B virus (HBV) infection is recognized globally as the leading cause of hepatocellular carcinoma (HCC). Research consistently highlights that certain patients maintain a persistent risk of developing HCC. This review is dedicated to exploring the mechanisms by which patients with HBV can develop HCC, even after years of viral suppression, and why these cases often have a worse prognosis compared to treatment-naive HBV patients. Understanding these dynamics is crucial for enhancing screening protocols and advancing towards a cure. However, the manuscript requires significant revisions before it can be considered for publication.

1.       The current manuscript includes only two figures, both of which are sourced from other articles. It is imperative for the authors to create original figures to better represent and support the unique insights of this review.

2.       The manuscript intends to address the worsened prognosis in HBV patients who develop HCC, yet this topic is not sufficiently covered. There is substantial literature reporting on this aspect (e.g., publications in Hum Cell. 2023 Sep;36(5):1773-1789. doi: 10.1007/s13577-023-00945-z ; Int J Mol Sci. 2023 Feb 7; 24(4):3263. doi: 10.3390/ijms24043263; Cancers (Basel). 2022 Nov 24;14(23):5781. doi: 10.3390/cancers14235781; J Hepatol. 2022 Jan;76(1):148-159. doi: 10.1016/j.jhep.2021.08.029; Pharmacol Res. 2020 Jul;157:104800. doi: 10.1016/j.phrs.2020.104800; Infect Agent Cancer. 2021 Mar 23;16(1):20. doi: 10.1186/s13027-021-00359-2; Transl Oncol. 2018 Apr;11(2):511-517. doi: 10.1016/j.tranon.2018.02.015; Cancer Lett. 2014 Oct 1;352(2):245-52. doi: 10.1016/j.canlet.2014.07.004) . Therefore, it is essential to incorporate a detailed discussion on the role of HBV in the drug resistance of HBV-associated HCC to provide a comprehensive view of the subject matter.

3.       The language of the manuscript necessitates meticulous proofreading and editing by a native English speaker. This will ensure clarity, fluency, and precision in the presentation of the research, making it accessible and comprehensible to a global academic audience.

Comments on the Quality of English Language

   The language of the manuscript necessitates meticulous proofreading and editing by a native English speaker. This will ensure clarity, fluency, and precision in the presentation of the research, making it accessible and comprehensible to a global academic audience.

Reviewer 3 Report

Comments and Suggestions for Authors

Interesting review on a cutting-edge topic.

In the paragraph on HCC risk factors, the authors should speculate more on the rising incidence of NASH/NAFLD-related HCC and on methods to prevent it, for example bariatric surgery (in this regard cite the recent SRMA: PMID: 33721336)

There is evidence on the comparison between different preventive action against HCC of several antiviral therapy .....maybe a table summarizing the relevant studies could be useful to the reader

Round 2

Reviewer 3 Report

Comments and Suggestions for Authors

THe revised version of the manuscript is OK. Thank you!